# Research on the Change in Chemical Composition and Fungal Resistance of Moso Bamboo with Heat Treatment

**DOI:** 10.3390/polym15020453

**Published:** 2023-01-15

**Authors:** Wangwang Yu, Yong Wang

**Affiliations:** School of Mechanical Engineering, Nanjing Vocational University of Industry Technology, Nanjing 210023, China

**Keywords:** bamboo, thermal modification, EMC, anti-mildew property

## Abstract

Bamboo, as a potential alternative to biomass materials, has gained more attention from the bamboo manufacturing industry in China. However, the drawbacks, such as the dimensional instability, and low antifungal and hydrophilic properties of bamboo, inhibit its application and shorten its service life. The objective of this work was to analyze the change in chemical components and fungal resistance of moso bamboo with heat treatment. For this objective, moso bamboo specimens were thermally modified in an oven at different temperatures for a fixed duration. The results showed that the parenchymal cells distorted and deformed in comparison to the control after thermal modification. After thermal modification in an oven, the crystallinity index increased from 39% to 53%. Owing to the thermal modification, the hemicellulose and cellulose relative content decreased, as confirmed by FTIR and XPS analysis. Thus, the dimensional stability and antifungal properties of the thermally modified bamboo specimens improved.

## 1. Introduction

Bamboo, as a potential alternative to biomass materials, has gained attention from the bamboo manufacturing industry in China [1]. Notably, bamboo has many advantages such as being lightweight, having excellent mechanical strength, a short growth cycle, and a wide application in the fields of building façades, furniture, construction, and decoration [2]. Unfortunately, the inner drawbacks, such as the dimensional instability, low antifungal property, and hydrophilic property of bamboo, inhibit its application and shorten its service life [3]. Over the years, many efforts have been made to improve the dimensional stability, anti-mildew property, and hydrophilic property of bamboo and bamboo-based products [4]. Thermal modification is considered to be a cost-effective, sustainable chemical, and ecofriendly approach to improving the abovementioned drawbacks of bamboo.

Bamboo tubes consist of cellulose, lignin, hemicellulose, ash, and other compositions. In addition, these chemical compositions contain amorphous phases. When exposed to higher temperature conditions, these chemical compositions exhibit viscoelastic and plastic behaviors. Compared with the traditional heat treatment medium, saturated steam, dimethyl silicone oil, tung oil, water, etc., bamboo modified by an oven will not produce a large amount of waste solution, and in the process of treatment can be modified to improve the dimensional stability and mildew resistance [5]. Additionally, chemical agent impregnation can effectively enhance the mechanical properties and anti-mildew properties of bamboo tubes. Unfortunately, the utilization of chemical agents is not environment-friendly and can be harmful to the human body when these bamboo-based products are applied in our daily life. Not much literature has systematically analyzed the change in chemical components and fungal resistance of moso bamboo from the chemical functional groups’ standpoint and microstructural level. Furthermore, the literature on the antifungal property of moso bamboo is rare compared to that of wood [6]. Traditional studies have been limited to the effects of the modification methods on the macroscopic physical and mechanical properties of bamboo or wood, such as the equilibrium moisture content, density, tensile strength, compressive strength, surface hardness, and flexural strength [7,8,9]. The heat treatment reduces the hygroscopic property of the bamboo, consequently reducing its shrinkage and swelling properties and improving its dimensional stability when the treatment is above 150 °C. At 180 °C or higher, heat treatment can significantly improve the anti-fungi property of bamboo. However, our research on the modification should focus on the enhancement of the practical applications of bamboo and bamboo products, such as their antifungal properties, dimensional stability, and hydrophobic properties. Furthermore, the study of the relationship between the change in specific chemical functional groups and the change in mold resistance is beneficial for our further understanding of the thermal modification mechanism [10]. X-ray photoelectron spectroscopy (XPS), Fourier transform infrared (FTIR), and scanning electron microscopy (SEM) are modern techniques for material surface interface characterization, which can qualitatively or quantitatively analyze the specific changes in carbon and oxygen content, as well as observe the changes in the bamboo cell wall, thus fully revealing the thermal modification mechanism [11]. Additionally, for outdoor application, the fungal-resistance properties of the treated bamboo specimens are also important. Therefore, the fungal -resistance properties of the untreated bamboo and treated bamboo specimens were also analyzed in this paper [12].

The objective of this work was to analyze the change in chemical components and fungal resistance of moso bamboo with heat treatment. For this objective, moso bamboo specimens were thermally modified in an oven at different temperatures and a fixed duration. At the same time, the micromorphology, EMC, oven density, chemical functional groups, and antifungal properties of the untreated and treated bamboo specimens were analyzed in detail by means of SEM, XPS, FTIR, and a physical property test.

## 2. Materials and Methods

### 2.1. Sample Preparation

Four-year-old moso bamboo was collected from Qingyuan, China. Moso bamboo was cut from 1 m (height from base). There were no significant defects on the bamboo surfaces. The bamboo culms were split into average dimensions of 100 mm × 5 mm × 5 mm (length × width × thickness). Then, the bamboo strips were thermally modified in an oven at different temperatures and a fixed duration. In this study, sample A represented the untreated bamboo samples. Sample B, sample C, and sample D represented the bamboo samples treated at 160, 170, and 180 °C, with a fixed treatment duration of 30 min.

### 2.2. Scanning Electron Microscopy

Firstly, the bamboo samples were cut into average sizes of 5 mm × 5 mm × 1 mm (length × width × thickness). Then, the surface of the bamboo samples was polished by a knife until the surface became smooth and flat. Lastly, an FEI Quanta 200 Scanning electron microscope (SEM) was used to study the change in the micromorphology of the control and treated bamboo specimens.

### 2.3. Measurement of the Main Chemical Compositions of the Bamboo

The change in the main chemical compositions of the bamboo after thermal modification was investigated by the methods reported by the U.S. National Renewable Energy Laboratory (NREL) [13]. Firstly, a UV-vis spectrometer (1200 series, Agilent, Jinghong, Shanghai, China) was used to analyze the acid-soluble lignin fraction in the solution. Additionally, the xylose, glucose, and arabinose in the solution were analyzed through high-performance liquid chromatography (2500 series, Quanta, Sanqiang, Shanghai, China). The relative content of lignin and hemicellulose can be obtained from the wet chemistry method; the relative content of cellulose was obtained from the known hemicelluloses and lignin content. The determination process was repeated three times with an average of 0.1% accuracy for each specimen.

### 2.4. XPS

Firstly, the bamboo samples were ground into a powder, and we ensured they could pass through a 200-mesh sieve. Before the XPS test, the untreated and treated bamboo powders were dried in a heating chamber at 103 degrees C for 24 h. X-ray photoelectron spectroscopy (XPS) was used to measure the C1s and O1s of the untreated and treated bamboo samples. The XPS system was equipped with an AL Kα X-ray source. The binding energy range of the untreated and treated bamboo samples was from 0 to 1350 eV. The determination process was repeated three times

### 2.5. Fourier Transform Infrared (FTIR)

Firstly, the bamboo samples were ground into a powder, and we ensured they could pass through a 200-mesh sieve. Before the FTIR test, the untreated and treated bamboo powders were dried in a heating chamber at 103 degrees C for 24 h. Then, the control and treated bamboo specimens were ground into a bamboo powder and investigated by Fourier transform infrared (FTIR) spectroscopy. The range of the wavenumber was 4000–400 cm^−1^ at a scanning speed of 40 cm^−1^/s. The determination process was repeated three times.

### 2.6. Anti-Mildew Property Test

First, we weighed 46 g of glucose potato agar (PDA) powder and dissolved the glucose potato agar powder in 1000 g of deionized water. Then, we stirred the mixed solution with a magnetic stirrer for 40 min until the potato dextrose agar powder was completely dissolved in the water. Thus, 1000 mL of PDA medium solution was obtained. The PDA solution was then poured into three 500 mL flasks. The three flasks sealed with waterproof paper were sterilized in a pressure-steam sterilizer at 121 °C for 1 h. After that, the solution in the flasks was poured into the petri dishes, waiting for its solidification. The untreated bamboo specimens were used as the control group according to the Chinese National Standard (GB/T 18261–2013). Then, we poured the sterilized PDA into Petri dishes and waited for it to cure at 15 degrees C. Finally, we inoculated the PDA with *Aspergillus niger* and then placed the bamboo specimens to be tested in a Petri dish. During the experiment, we recorded the changes in the bamboo sample surface every 2 days and took photos of the changes with a camera.

### 2.7. Statistical Analysis

Tukey’s tests were used to distinguish the differences between the different bamboo samples via statistical software SPSS. Additionally, 12 replicates were used to measure means for modulus of elasticity, modulus of rupture, tensile strength, and oven-density. Different capital letters represent the significant difference between different groups (*p* < 0.05).

## 3. Results and Discussion

### 3.1. Micromorphology Analysis

The change in the micromorphology of the bamboo directly affected the macrophysical property of the biomass materials. Thus, the micromorphology of the bamboo specimens after thermal modification were analyzed by SEM, and the related results are presented in Figure 1. Figure 1(A(1)–A(3)) displays the microstructure of the untreated bamboo, exhibiting a natural porous structure with plump and round parenchymal cells. After thermal modification, as shown in Figure 1(B(1)–B(3)), the parenchymal cells were distorted and deformed in comparison to the control. A similar observation was obtained by the previously reported literature [14]. The decomposition of the chemical components in the bamboo cell walls and the evaporation of the moisture in the high-pressure and high-temperature environment led to the thin-walled cells’ deformation during the thermal modification [15]. The detailed changes in the chemical components of the bamboo inner tissue were confirmed by means of the FTIR, XPS, and physical property tests.

### 3.2. XRD Analysis

XRD was used to identify the crystallinity index degree between the untreated bamboo and the thermally modified bamboo (Figure 2A,B). The XRD curves of the bamboo samples presented two main diffraction peaks at 15.6° and 21.4°, respectively. The crystallinity index degree of the untreated bamboo and thermally modified bamboo can be calculated according to Segal’s formula [16]. The crystallinity index of the control was 39%. After thermal modification in an oven, the crystallinity index increased from 39% to 53%. When the temperature was above 140 degrees C, the amorphous zone of the hemicellulose began to degrade [17]. This was because the hemicellulose more easily hydrolyzed than the cellulose and lignin [18]. Additionally, the degradation of the cellulose amorphous content also led to the increase in the crystallinity index of the thermally modified bamboo sample.

### 3.3. Chemical Components Analysis

The main chemical components of the untreated and treated bamboo samples after heat treatment are presented in Table 1. When the treatment temperature was above 160 degrees C, the relative content of the hemicellulose and cellulose decomposed significantly. For example, the relative content of hemicellulose decreased from 20.5% to 18.1%, indicating that the hemicellulose content began to decrease at the high treatment temperature. According to the previous literature [19], the hemicellulose was less stable due to the branched-chain structure and lower molecular weight, and thus was easier to decompose in comparison to the cellulose and lignin [20]. Additionally, the lignin relative content rose, while the relative content of the hemicellulose and cellulose reduced compared with that of the control [21]. On the one hand, the lignin presented a better thermal stability than the cellulose and hemicellulose. On the other hand, the lignin condensation and crosslinking reactions also positively contributed to the lignin content [22].

### 3.4. FTIR Analysis

The FTIR curves of different bamboo samples are shown in Figure 3. The region between 4000 and 2000 cm^−1^ was dominated by the stretching vibrations of the hydrogen-containing groups and the triple-bond and cumulative double-bond regions. The stronger absorption peaks at 3420 and 2917 cm^−1^ were the stretching vibrations of the -CH and -OH, respectively, which were caused by the action of the high temperature and steam; the hydroxyl groups became active and condensed with each other to remove the water molecules to form ether bonds, thus reducing the number of free hydroxyl groups. But due to the hygroscopic nature of the KBr pressed tablets, the water could not be completely removed, which affected the determination of the peak intensity. The region between 2000 cm^−1^ and 800 cm^−1^ provided more information on the chemical composition of the bamboo before and after treatment. The highest peak at 1732 cm^−1^ was due to the ester linkage of the acetyl and ester or carboxyl groups in the hemicellulose and the coumaric acid in lignin, and this nonconjugated carbonyl peak decreased with the increasing temperature and time due to the degradation of the hemicellulose. The absorption peaks at 1602 and 1507 cm^−1^ did not change significantly, indicating that the benzene ring skeleton in lignin was relatively stable and did not change significantly. The change in the intensity of the absorption peak at 1376 cm^−1^ was due to the stretching vibration of the C-H and C-H on the phenolic hydroxyl groups in the hemicellulose and cellulose. The weakening of the absorption peak at 1043 cm^−1^ indicated a decrease in the number of C-O bonds in the cellulose and hemicellulose.

### 3.5. XPS Analysis

X-ray photoelectron spectroscopy (XPS) is a modern technique for material surface interface characterization, which can qualitatively or quantitatively analyze the specific changes in the carbon and oxygen content, as well as show the changes in the bamboo cell wall, thus fully revealing the thermal modification mechanism. As shown in Figure 4 and Table 2, after thermal modification, the C_1_ (C-C or C-H) composition increased from 36.5% to 49.6%, while the C_2_ (C-O), C_3_ (C=O and O-C-O bonds) presented a decreasing tendency. According to the previous literature [23], the C_2_ and C_3_ composition belongs to hemicellulose and cellulose, and the decrease in the C_2_ and C_3_ components confirmed the decomposition of hemicellulose and cellulose. The change in O1s, shown in Table 2, also confirmed the change in the wet chemistry method analysis. Therefore, the bamboo surface was low in hemicellulose and cellulose, while it was rich in lignin.

### 3.6. EMC Analysis

EMC, as a physical index, can effectively evaluate the dimensional stability of samples after thermal modification. The results of the EMC of the control and treated bamboo samples are shown in Figure 5. The EMC of the control was 15.6%. The EMC of the treated bamboo samples presented a decreasing tendency. The results from the statistical analysis showed that the EMC of the treated bamboo samples significantly decreased from 15.6% to 7.9%. According to the reported literature, hemicellulose contains a large number of hydroxyl groups on its surface. It is well known that hydroxyl groups are hydrophilic and can absorb large amounts of water in air [24]. However, the hydroxyl groups decrease with the degradation of hemicellulose [25]. Thus, the equilibrium moisture content of the treated bamboo samples tends to decrease, which indicates the improvement in the dimensional stability of the specimens [26].

### 3.7. Mold-Resistant PROPERTY Analysis

Figure 6 A,B shows the anti-mildew property and corresponding pictures. The bamboo and bamboo-based products are often influenced by mildew, thus decreasing their service life and mechanical properties. Therefore, it was of importance to investigate the anti-mildew property of the untreated and treated bamboo specimens through incubating the bamboo specimens with *A. niger*. From Figure 6A,B, we can see that the infection ratio of the control quickly increased to 100% after 10 days. Fortunately, the antifungal property of the thermally modified bamboo samples was enhanced. Namely, the heat treatment positively contributed to the antifungal property of the bamboo [27]. For instance, the infection ratio of the treated bamboo increased to 100% in 16 days after the treatment temperature was set at 160 degrees C. When the treatment temperature increased to 180 degrees C, the treated bamboo specimens exhibited an excellent antifungal property in comparison to that of the control [28]. During the heat treatment process, the internal nutrients of the bamboo, such as the polysaccharides and starch, were further reduced as the thermal modification temperature increased [29]. The reduction in a large number of hydrophilic groups leads to an increase in the hydrophobicity of the bamboo surface, which reduces the adhesion of *A. niger* to the bamboo surface [30]. On the other hand, the increased hydrophobicity of the bamboo material also makes it difficult for the nutrients inside the bamboo material to be transferred to the outside, which is detrimental to the growth of *A. niger* [31]. Therefore, the anti-mildew property of the thermally modified bamboo increased.

## 4. Conclusions

In this study, the change in the chemical composition and fungal resistance of moso bamboo with heat treatment was analyzed by means of XPS, XRD, FTIR, and the wet chemistry method. The results showed that 180 °C and 30 min should be the optimal treatment parameters. SEM was used to analyze the micromorphology of the untreated and treated bamboo samples. The results further showed that the parenchymal cells distorted and deformed in comparison to the control. The decomposition of the chemical components in the bamboo cell walls and evaporation of moisture in the high-pressure and high-temperature environment led to the thin-walled cells’ deformation during the thermal modification. The hemicellulose and cellulose decreased, as confirmed by the wet chemistry method. The related peak and C1s of the hemicellulose and cellulose decreased as shown in the FTIR and XPS analysis. The EMC of the treated bamboo samples significantly decreased from 15.6% to 7.9% due to the decomposition of the hydroxyl groups in the hemicellulose. Thus, the dimensional stability of the bamboo samples was enhanced. Lastly, the thermal modification improved the antifungal property of the treated bamboo samples. This work provides a simple heat treatment method for improving the mildew resistance of bamboo products.

## Figures and Tables

**Figure 1 polymers-15-00453-f001:**
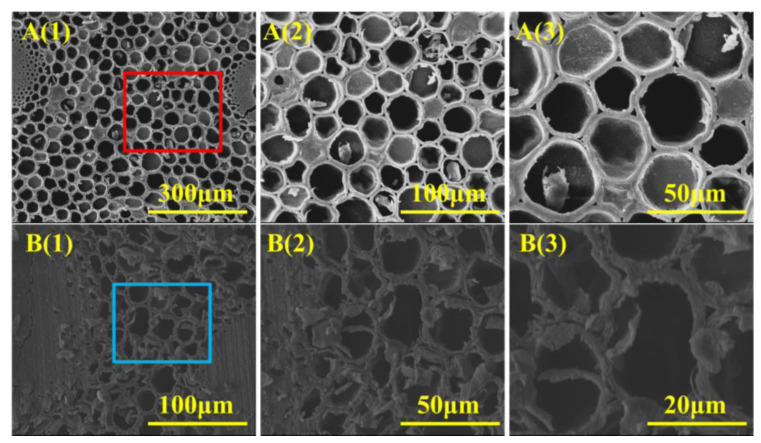
SEM images of the cross-section surface structure of Sample A (**A**(**1**)–**A**(**3**)) and Sample B (**B**(**1**)–**B**(**3**)). The different color boxes is the observation area.

**Figure 2 polymers-15-00453-f002:**
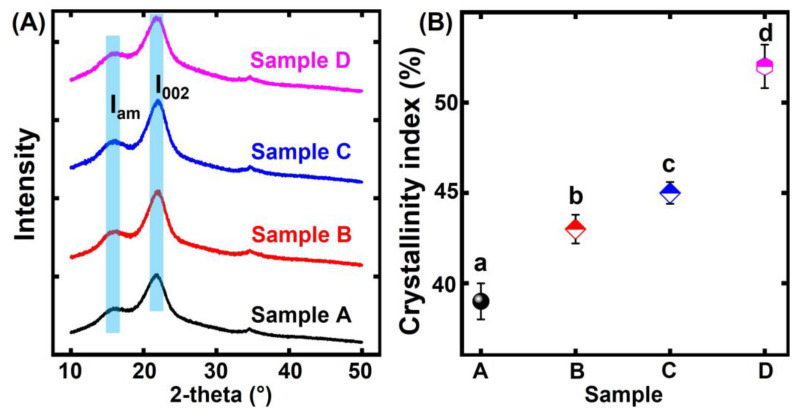
XRD spectra of moso bamboo: (**A**) related XRD curves of different bamboo specimens; (**B**) crystallinity index of different bamboo specimens. Different significant differences between different groups (*p* < 0.05). The error bar in the picture represents the standard deviation.

**Figure 3 polymers-15-00453-f003:**
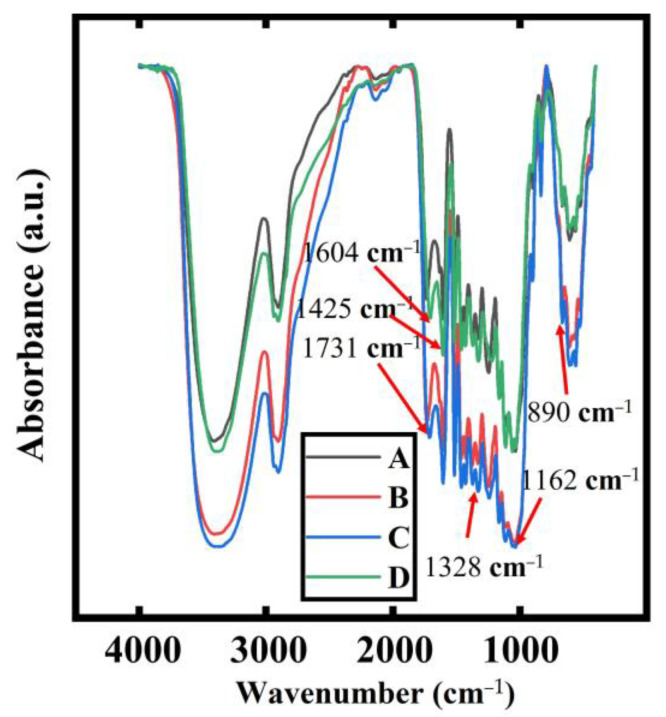
FTIR spectra of the moso bamboo.

**Figure 4 polymers-15-00453-f004:**
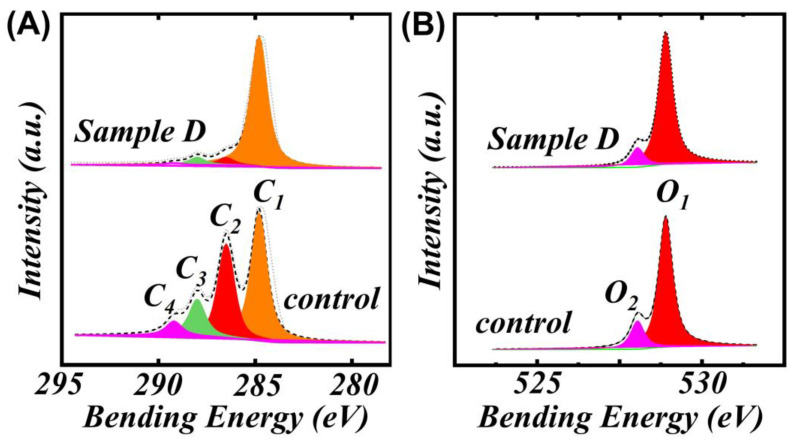
XPS curves of the different bamboo samples. (**A**) C1s of different bamboo specimens; (**B**) O1s of different bamboo samples.

**Figure 5 polymers-15-00453-f005:**
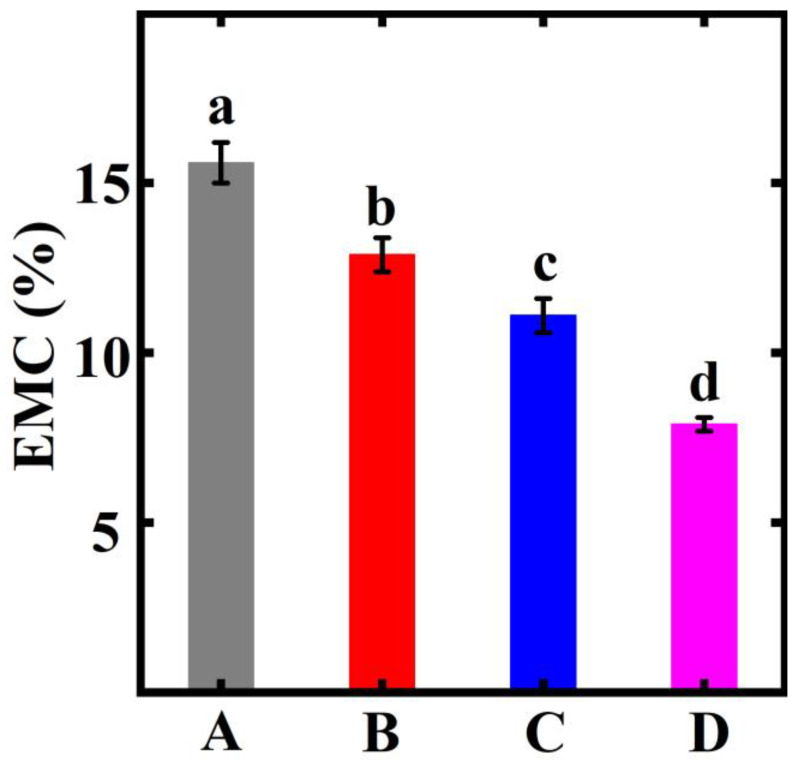
Physical property (EMC) of the moso bamboo. Different significant difference between different groups (*p* < 0.05). The error bar in the picture represents the standard deviation.

**Figure 6 polymers-15-00453-f006:**
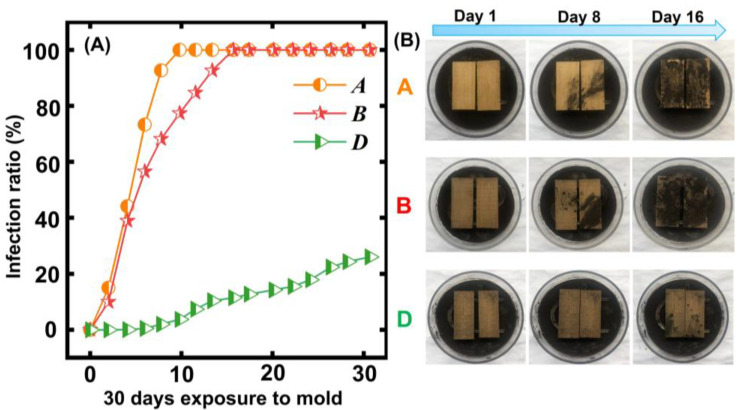
The results of the different bamboo samples exposed to mold for 30 days: (**A**) the infection ratio of different bamboo samples in one month; (**B**) corresponded picture during the anti-fungal test.

**Table 1 polymers-15-00453-t001:** The chemical compositions of the untreated and thermally modified bamboo specimens.

Sample	Hemicellulose (%)	Cellulose (%)	Lignin (%)
Sample A	20.5 (0.1)	46.9 (0.1)	24.5 (0.1)
Sample B	19.6 (0.1)	45.7 (0.2)	26.6 (0.2)
Sample C	18.9 (0.2)	44.8 (0.1)	28.5 (0.2)
Sample D	18.1 (0.3)	42.6 (0.3)	31.3 (0.3)

**Table 2 polymers-15-00453-t002:** Specific changes in the carbon and oxygen content of the thermally modified bamboo samples.

Specimen	O/C	C_1_	C_2_	C_3_	C_4_	O_1_	O_1_
Sample A	0.55	36.5	51.3	10.5	1.7	93.8	6.2
Sample D	0.31	49.6	40.3	8.9	1.2	92.8	7.2

## Data Availability

Data will be available on request.

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
