# Peer review of "Research on the Change in Chemical Composition and Fungal Resistance of Moso Bamboo with Heat Treatment"

_polymers, 2023, doi:10.3390/polym15020453_

Round 1

Reviewer 1 Report

There have been many studies on the durability of bamboo heat treatment. There is no innovation in the methods and conclusions of this study. It is also consistent with the previous research conclusion that heat treatment leads to the reduction of polysaccharide components such as hemicellulose and the enhancement of durability.

Author Response

Response: Thanks again to the reviewer on suggesting to properly address the significance of the work. As we known, bamboo tubes consist of cellulose, lignin, hemicellulose, ash and other compositions. In addition, these chemical composition contains amorphous phases. When exposed to higher temperature condition, these chemical compositions exhibit viscoelastic and plastic behaviours. Compared with the traditional heat treatment medium, saturated steam, dimethyl silicone oil, tung oil, water, and so on, etc., bamboo modified by an oven will not produce a large amount of waste solution, and in the process of treatment can be modified to improve the dimensional stability and mildew -resistance of bamboo. In addition, chemical agent impregnation can effectively enhance the mechanical properties and anti-mildew properties of bamboo tubes. Unfortunately, the utilization of chemical agents is not environment-friendly and can be harmful to the human body when these bamboo-based products are ap-plied in our daily life. NThere are ot muchlittle literature hasve systematically ana-lyzed the change in chemical components and fungal i-resistance of moso bamboo from the chemical functional groups’ standpoint and the micro-structurale level. In addition, the literature on thef anti-fungal property of moso bamboo is rareely been reported compared towith that of the wood. Traditional studies have been limited to the effects of the modification methods on the macroscopic physical and mechani-cal properties of bamboo or wood, such as the equilibrium moisture content, density, tensile strength, compressive strength, surface hardness, and flexural strength. The heat treatment reduces the hygroscopic property of the bamboo, consequently reducing the hygroscopic property of the bamboo, thus reducing its shrinkage and swelling properties and improving the dimensional stability when the treatment is above 150 °C. At 180°C or higher, heat treatment can significantly improve the anti‐fungi property of bamboo.However, our research on the modification should focus on itsthe enhancement of the modification method thefor practical applications of bamboo and bamboo -products, such as their anti-fungal propertiesy, dimensional stability, and hydrophobic properties. Furthermore, the study of the relationship between the change in specific chemical functional groups and the changes in mold resistance is beneficial forto our further understanding of the thermal modification mechanism. X-ray photoelectron spectroscopy (XPS), Fourier transform infrared (FTIR), and sScanning electron microscopy (SEM) are modern techniques for material surface in-terface characterization, which can qualitatively or quantitatively analyze the spe-cific changes in carbon and oxygen content, as well as observe the changes in the bamboo cell wall, thus fully revealing the thermal modification mechanism. Ad-ditionally, for outdoor application, the fungal-resistance property of the treated bam-boo specimens are also important. Therefore, the fungal -resistance property of the untreated bamboo and treated bamboo specimens were also analyzed in this paper. This work provides a simple heat treatment method for improving the mildew resistance of bamboo products. In our furture work, we want focus on the change in mechanical property of bamboo cell wall.

Reviewer 2 Report

The topic of the research work and manuscript is really interesting and provides new information. However there are several issues to be addressed towards its quality improvement before publication.

In line 8, the "has been gain" needs improvement. In line 13, "an" should be changed to "a". "Hemicellulose" should be used in plural form due to the variety of their nature. Please, provide more key words, to ensure readability for example: chemical composition, biological durability, hygroscopicity, fungi etc. In line 24, there is a type error. In line 24, the word "property" should be changed to "strength". In line 30, 56 and 69, "an" should be altered to "a". In line 36, the "There are little literature have" needs imrovement. In introduction, use the relevant article of  https://doi.org/10.5552/drvind.2021.2026 to support a brief reference concerning the changes induced in chemical composition and structure of wood after thermal modification. In my opininon, that would enrich your state-of-the-art analysis and be helpful to the reader to comprehend what is the background knowledge. Are there any similar previous studies on thermal modification of bamboo analyzing the changes in its chemical properties? If yes, you should include the paper in the introduction theroretical approach of the topic. In line 56, "thermal" rather be changed to "thermally". In line 63, you should talk in past tense, not present. In line 69, degree should be changed to the respective symbol. In materials and methods, you do not refer why did you choose the specific temperatures and duration of treatment? where did you base these choices of conditions? Was the oven pre-heated? Which was the atmosphere inside the oven? From the 30min of treatment duration, which was the time needed for the temperature to stabilize in the oven? You rather change oven to "heating chamber". What happened after the completion of treatment? Which was the next step/steps? Has the dimensions of bamboo strip specimens something to do with the intensity of treatment? In line 73, the word "waited" should be altered to "intended". Which was the model and details (manufacturer, country etc.) of "Liquid Chromatography" equipment? (as well as for the rest of equipment used in this experimental work). The phrase of 84-85 needs to be clarified, what do you mean can be analyzed after the experiment? In line 89 and 96, the "in a Oven" needs improvement. How many specimens were tested for each property? Be careful with the symbols that need to have superscripts. You rather use passive voice in descriptions of methodology applied. In line 102, add a "d" in "deionize". Were the Petri dishes sterilized? visual assessment of surface? Were the specimens sterilized?Did you record as well the mass loss or only the Scientific names of species should be written in italics. In 108 line the "with our phones" is not scientific statement needs improvement. In which standard did you base this experimental methodology? You did not refer anything about statistical analysis of the results (if any).  Figure 4 can be extended a little bit to be more readable. In figure 7 caption the word bamboo is referred twice. In 244 line, "was" should be added to "enhanced". How could the specific conditions used in this study be considered in the view of effieciency? Which were the optimal ones? how could the duration be considered (satisfied?needs perhaps extension? should be further researched?). Highlight the meaning and significance of this work, especially in the end of manuscript in conclusions for the future utilization of bamboo biomass.  

Author Response

Response to reviewer 2

Point 1: In line 8, the "has been gain" needs improvement.

Response: Thanks again to the reviewer on suggesting to properly address the significance of the work. We have revised that in our revised version.

Point 2: In line 13, "an" should be changed to "a". "Hemicellulose" should be used in plural form due to the variety of their nature. Please, provide more key words, to ensure readability for example: chemical composition, biological durability, hygroscopicity, fungi etc.

Response: Thanks again to the reviewer on suggesting to properly address the significance of the work. The “an” has been revised into “a” in our revised version. As you will see, in the revised version, we have carefully taken the comments into consideration in preparing our revision. We have corrected that in the revised version.

Point 3:In line 24, there is a type error.

Response: Thanks again to the reviewer on suggesting to properly address the significance of the work. The type error has been revised in our revised version. We have corrected that in the revised version.

Point 4:In line 24, the word "property" should be changed to "strength". In line 30, 56 and 69, "an" should be altered to "a".

Response: Thanks again to the reviewer on suggesting to properly address the significance of the work. The word “property” has been changed to “stength”.

Point 5:In line 36, the "There are little literature have" needs imrovement. In introduction, use the relevant article of  https://doi.org/10.5552/drvind.2021.2026 to support a brief reference concerning the changes induced in chemical composition and structure of wood after thermal modification. In my opininon, that would enrich your state-of-the-art analysis and be helpful to the reader to comprehend what is the background knowledge. Are there any similar previous studies on thermal modification of bamboo analyzing the changes in its chemical properties? If yes, you should include the paper in the introduction theroretical approach of the topic.

Response: Thanks again to the reviewer on suggesting to properly address the significance of the work. The relevant article of https://doi.org/10.5552/drvind.2021.2026 has been added in our revised version.

Point 6:In line 56, "thermal" rather be changed to "thermally".

Response: Thanks again to the reviewer on suggesting to properly address the significance of the work. “thermal” has been changed into “thermally”.

Point 7:In line 63, you should talk in past tense, not present. In line 69, degree should be changed to the respective symbol. In materials and methods, you do not refer why did you choose the specific temperatures and duration of treatment? where did you base these choices of conditions? Was the oven pre-heated? Which was the atmosphere inside the oven? From the 30min of treatment duration, which was the time needed for the temperature to stabilize in the oven? You rather change oven to "heating chamber". What happened after the completion of treatment? Which was the next step/steps? Has the dimensions of bamboo strip specimens something to do with the intensity of treatment?

Response: Thanks again to the reviewer on suggesting to properly address the significance of the work. In line 63, we have used past tense in our revised version. Degree has been changed in the respective symbol. We has explained why we choose the temperatures in our introduction section. We have changed the “oven” into “heating chamber”. There is no relationship between dimensions of bamboo strip specimens and intensity of treatment.

Point 8:In line 73, the word "waited" should be altered to "intended". Which was the model and details (manufacturer, country etc.) of "Liquid Chromatography" equipment? (as well as for the rest of equipment used in this experimental work).

Response: Thanks again to the reviewer on suggesting to properly address the significance of the work. The model and details (manufacture, country etc) of “liquid chromatography” equipment has been added in our revised version.

Point 9:The phrase of 84-85 needs to be clarified, what do you mean can be analyzed after the experiment?

Response: Thanks again to the reviewer on suggesting to properly address the significance of the work. The relative content of lignin and hemicellulose can be obtained from wet chemistry method, the relative content of cellulose was obtained from the known hemicelluloses and lignin content. The determination process was repeated three times and average to 0.1 % accuracy for each specimen. We have corrected that in the revised version.

Point 10:In line 89 and 96, the "in a Oven" needs improvement. How many specimens were tested for each property? Be careful with the symbols that need to have superscripts. You rather use passive voice in descriptions of methodology applied.

Response: Thanks again to the reviewer on suggesting to properly address the significance of the work. We have changed the “oven” into “heating chamber”. The determination process was repeated three times.

Point 11:In line 102, add a "d" in "deionize". Were the Petri dishes sterilized? visual assessment of surface? Were the specimens sterilized?Did you record as well the mass loss or only the Scientific names of species should be written in italics.

Response: Thanks again to the reviewer on suggesting to properly address the significance of the work. We have add a “d” in “deionize”. Thus, 1000mL of PDA medium solution was obtained. The PDA solution was then poured into three 500mL flasks. The three flasks sealed with waterproof paper were sterilized in a pressure-steam sterilizer at 121 °C for 1h. After that, the solution in the flasks was poured into the petri dishes, waiting for its solidification. The untreated bamboo specimens were used as the control group according to the Chinese National Standard (GB/T 18261–2013). We did not record the mass loss in our experiment.

Point 12: In 108 line the "with our phones" is not scientific statement needs improvement. In which standard did you base this experimental methodology? You did not refer anything about statistical analysis of the results (if any). In figure 7 caption the word bamboo is referred twice.

Response: Thanks again to the reviewer on suggesting to properly address the significance of the work. We have revised that into with camera. The word bamboo referred twice in figure 7 has been revised. The statistical analysis has been added into our revised version.

Point 13: In 244 line, "was" should be added to "enhanced". How could the specific conditions used in this study be considered in the view of effieciency? Which were the optimal ones? how could the duration be considered (satisfied?needs perhaps extension? should be further researched?). Highlight the meaning and significance of this work, especially in the end of manuscript in conclusions for the future utilization of bamboo biomass.  

Response: Thanks again to the reviewer on suggesting to properly address the significance of the work. 180 and 30 min should be the optimal treatment parameters. “was” has been added to “enhanced” This work provides a simple heat treatment method for improving the mildew resistance of bamboo products. In our furture work, we want focus on the change in mechanical property of bamboo cell wall.

Round 2

Reviewer 1 Report

The authors had addressed all my comments. Therefore, the revised manuscript can be accepted for publication

Reviewer 2 Report

As I have checked the authors have implemented the proposed changes in the revised verion of manuscript towards the improvement of their work. Almost all the changes have been implemented and in my opinion, the manuscript is well-prepared and organized enough to be accepted for publication in this journal. I remain at your disposal for any clarification.